

# Qualitative study of healthcare providers' current practice patterns and barriers to successful rehydration for pediatric diarrheal illnesses in Kenya

Darlene R. House[1,2], Philip Cheptinga[2,3], Daniel E. Rusyniak[1] and Rachel C. Vreeman[2,4]

[1] Department of Emergency Medicine, Indiana University School of Medicine, Indianapolis, IN, United States of America
[2] Academic Model Providing Access to Healthcare (AMPATH), Eldoret, Kenya
[3] Department of Child Health and Paediatrics, Moi University School of Medicine, Eldoret, Kenya
[4] Department of Pediatrics, Indiana University School of Medicine, Indianapolis, IN, United States of America

## ABSTRACT

**Background**. For children worldwide, diarrhea is the second leading cause of death. These deaths are preventable by fluid resuscitation. Nasogastric tubes (NGs) have been shown to be equivalent to intravenous fluids for rehydration and recommended by the World Health Organization (WHO) for use in severe dehydration. Despite this, NGs are rarely used for rehydration in Kenya. Our objective was to evaluate clinicians' adherence to rehydration guidelines and to identify barriers to the use of NGs for resuscitating dehydrated children.

**Methods**. A case-based structured survey was administered to pediatric care providers in western Kenya to determine their choices for alternative rehydration therapies when oral rehydration and intravenous fluids fail. Providers then participated in a qualitative, semi-structured interview to identify barriers to using nasogastric tubes for rehydration. Analysis included manual, progressive coding of interview transcripts to identify emerging central themes.

**Results**. Of 44 participants, only four (9%) followed WHO guidelines that recommend quickly switching to NG for rehydration in their case responses. Participants identified that placing intravenous lines in dehydrated children is a challenge. However, when discussing NG use, many believed NGs are not effective for rehydration. Other participants' concerns surrounded knowledge and training regarding guidelines as well as not having NGs available.

**Discussion**. Healthcare providers in western Kenya do not report using NGs for rehydration in accordance with WHO guidelines for diarrheal illness with severe dehydration. Barriers to the use of NG tubes were lack of knowledge and availability. Education and implementation of guidelines using NG tubes for rehydration may improve outcomes of children suffering from diarrheal illness with severe dehydration.

Corresponding author
Darlene R. House, dhouse@iu.edu, dhouse@iupui.edu

## INTRODUCTION

Diarrhea is the second leading cause of death in children worldwide (*CDC, 2015*). It kills more children every year than AIDS, measles, and malaria combined (*CDC, 2015*). The majority of these deaths occur in developing countries where diarrheal illnesses are prevalent and access to healthcare is limited (*Kotloff et al., 2013*). Despite financial and resource limitations, these deaths are preventable by simple fluid resuscitation. Dehydrated children can successfully be treated by administering fluids intravenously, intraosseously, or through a nasogastric (NG) tube. Of these modalities, nasogastric tubes may be the most efficient in developing countries.

Nasogastric tubes have been shown to be equivalent to intravenous (IV) fluids for rehydrating children, even in moderate-to-severe dehydration (*Green, 1987*; *Hidayat et al., 1988*; *Gremse, 1995*; *Nager & Wang, 2002*; *Yiu, Smith & Catto-Smith, 2003*; *Fonseca, Holdgate & Craig, 2004*; *Oakley et al., 2010*; *Rouhani et al., 2011*; *Freedman et al., 2015*). Placement of NG tubes does not require advanced medical skills and can be employed by community health workers, nurses, or paramedics in various healthcare facilities to begin rehydration while preparing to send the child to a higher level of care. NGs are inexpensive, are relatively painless to insert, and can be placed rapidly. The Center for Disease Control and World Health Organization (WHO) recommend NG treatment for severe dehydration, especially when intravenous access is difficult to obtain (*CDC, 2003*; *UNICEF/WHO, 2005*).

Based on current literature evaluating rehydration practices in Kenya, providers primarily use oral and intravenous rehydration, while nasogastric tubes are only mentioned in treatment associated with malnutrition (*Blum et al., 2011*; *Talbert et al., 2012*). Consistent with our experience, despite effectiveness and role of NG tubes in rehydration guidelines, NGs are rarely used for rehydration in Kenya, where prevalence and mortality rates of diarrheal illness, similar to other sub-Saharan African countries, remain high (*Kotloff et al., 2013*; *UNICEF, 2015*). No literature pertaining to clinicians' adherence to WHO rehydration guidelines and why clinicians do not use nasogastric tubes was identified. Our objective was to evaluate clinicians' adherence to rehydration guidelines and to identify barriers to the use of nasogastric tubes in resuscitating dehydrated children.

## METHODS

### Study design

A qualitative study of pediatric care providers in western Kenya was performed from January 2014 to July 2014 to learn about current practice patterns for rehydration therapy and to identify barriers to successful rehydration and nasogastric tube use. The study was approved by both Indiana University Institutional Review Board (Approval IRB # 1209009483) and the Institutional Research and Ethics Committee at Moi Teaching and Referral Hospital (MTRH) (Approval #000936). Verbal consent was obtained from each participant prior to participation and recording of the survey and interview.

## Study setting

The study was conducted in the Uasin Gishu County, located in western Kenya with a population of 894,179 (*Kenya County Guide, 2017*). The study included all levels of Kenya's healthcare system, which is structured in a step-wise manner to provide appropriate level of care and resources to a wide range of healthcare needs. Dispensaries are the first level of care and provide outpatient treatment for simple conditions. Health centres are the next level of care providing more diagnostic testing with outpatient and inpatient services. Sub-district hospitals are similar but provide a wider range of surgical services. Within each county is also a district hospital with more comprehensive medical and surgical services. Complicated cases can be referred to a national referral hospital. Located in the Uasin Gishu County of western Kenya in Eldoret, MTRH serves as the national government referral hospital with sub-specialty services, emergency care, and comprehensive inpatient and outpatient services. Uasin Gishu County has 125 health facilities (one referral hospital, one district hospital, two sub-district hospitals, 23 health centres and 88 dispensaries) (*Kenya County Guide, 2017*).

Eldoret is also home to Moi University School of Medicine and serves as the base for the Academic Model Providing Access to Healthcare (AMPATH). AMPATH is an organization that provides HIV care, primary healthcare, and chronic disease management with clinic sites throughout western Kenya.

## Study population

Convenience sampling of pediatric healthcare providers in all levels of healthcare facilities in Uasin Gishu County was performed. Each of the referral, district, and two sub-district hospitals were included for interviews along with five health centres and four dispensaries for a total of 13 healthcare facilities. At each increasing level of care, there are more pediatric providers so a larger sample was included from those facilities.

Practitioners at various levels of training who care for children were chosen from all levels of health care facilities. All sites and participants were asked permission in person and shown a letter of approval for the study. No compensation was provided. The study included pediatric healthcare providers in western Kenya, including pediatricians, pediatric residents, medical officers, clinical officers, community health workers, and nurses who care for children. Participants were excluded if they did not provide care to children. The study included participants from various healthcare facilities, including health centres, district hospitals, and Moi Teaching and Referral Hospital in Eldoret, Kenya.

## Study protocol

A case-based structured survey was administered orally by a trained research assistant in a one-on-one setting with participants. The survey collected basic background information, including work setting (i.e., hospital vs clinic), position (i.e., physician, resident/registrar, nurse practitioner/clinical officer, nurse), and, for residents or faculty, what their medical specialty was. The survey also included a clinical scenario of a child presenting to the emergency department with clinical signs of severe dehydration. The child is unable to take oral fluids and attempts at placing an intravenous line have failed. In a structured format,

the participants were asked to classify the degree of dehydration and then state what mode of rehydration therapy they would use. If the participant chose to try either oral fluids or placing an intravenous line, they were informed that this method was unsuccessful. This continued until the participants chose an alternative therapy (either intraosseous, nasogastric tube, central line, venous cut down, or transferring the patient). Answers were audio recorded, transcribed, and compared to existing guidelines. The survey was provided in both English and Kiswahili depending on participant preference.

Once the clinician completed the survey portion of the study regarding their current practice patterns, a qualitative, semi-structured interview was conducted by the trained interviewer regarding barriers to using nasogastric tubes for rehydration therapy. An interview guide with open-ended questions was used to elicit responses regarding both attitudes surrounding resuscitation practice and the use of nasogastric tubes for rehydration therapy. Field notes were taken during interviews and immediately following them. The interviews were recorded and transcribed. All surveys and interview transcripts were de-identified.

### Analyses

Survey results were reported using descriptive statistics, including numbers and frequencies of answers given. For analysis of the key-informant interview portions of the study, inter-rater reliability was determined between independent reviewers' identification of codes and themes using Cohen's kappa coefficient. The open codes and extracted themes between investigators demonstrated a high degree of agreement with a Cohen's kappa coefficient of 0.72. Quality of the study is based on research process and empirical grounding of the findings.

Two investigators independently reviewed transcripts of the interview using constant comparative analysis to identify emerging themes (*Strauss, 1990*). This involved reading transcripts several times with line-by-line analysis of each transcript using open coding to elucidate meaning and processes. Axial coding was then performed to make connections between categories to organize themes into causal relationships. Hypotheses and concepts were then developed inductively from the data. A theoretical framework was developed that explained barriers to the use of nasogastric tubes by pediatric care providers for fluid resuscitation of severely dehydrated children. Dedoose software (*Dedoose, 2016*) was used for analyses. Select quotations from interviews were used to emphasize key themes.

To integrate further investigator triangulation, we had another reviewer read a subset of transcripts and compare their independent impressions and coding fit with primary reviewers' codes and themes.

## RESULTS

Forty-four providers participated in the survey and were subsequently interviewed (Table 1). Given the case scenario, the majority of practitioners classified the child as having severe dehydration ($n = 40$) while others classified the child as either mild ($n = 1$), moderate ($n = 2$) or in shock ($n = 1$). After being given the scenario of two unsuccessful intravenous attempts, the majority of providers chose to attempt another IV or to attempt giving oral

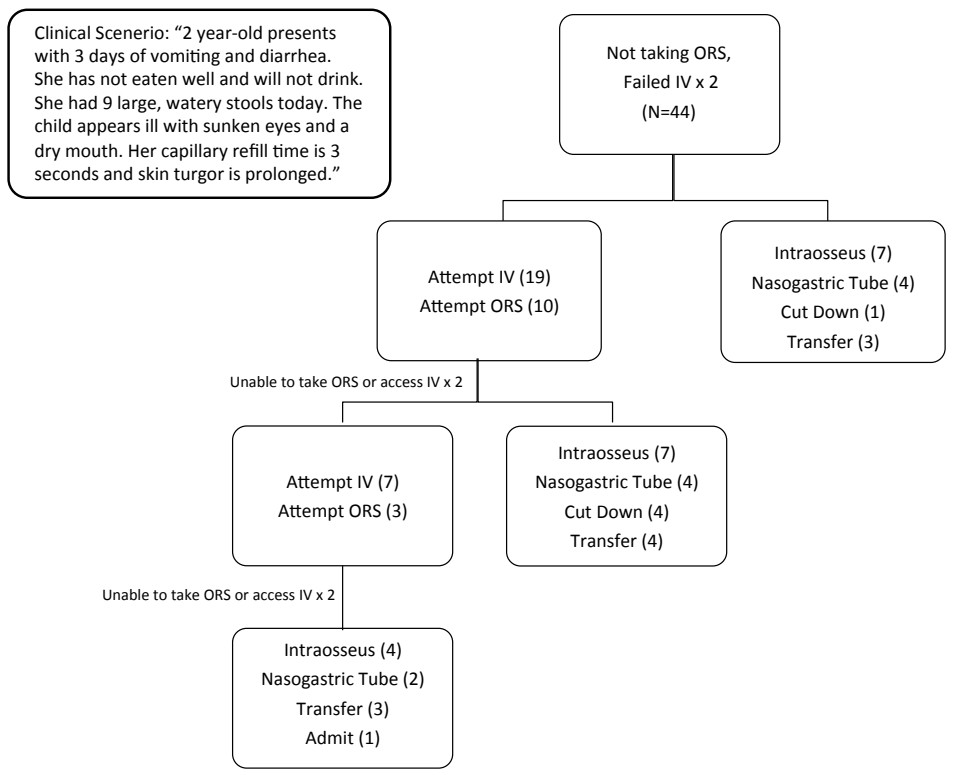

**Figure 1** Chosen alternative method for rehydration in case-based scenario.

**Table 1** Participant demographics.

|  | No.(%) of Participants ($N = 44$) |
|---|---|
| *Provider type* |  |
| Pediatrician | 3 (6.8) |
| Medical officer | 7 (15.9) |
| Clinical officer | 14 (31.8) |
| Nurse | 18 (40.9) |
| Community health worker | 2 (4.6) |
| *Practice location* |  |
| Referral Hospital | 20 (45.4) |
| District Hospital | 7 (15.9) |
| Sub-District Hospital | 8 (18.2) |
| Health centre | 5 (11.4) |
| Dispensary | 4 (9.1) |

rehydration salts (ORS). Only four participants (9%) adhered to WHO guidelines and chose NG as their next choice for rehydration (Fig. 1). After two more failures to place an IV or take ORS (a cumulative total of four unsuccessful IV or ORS attempts), a larger percentage of providers would try an NG (13%) but the majority (62%) still chose an invasive technique (i.e., IV, intraosseous, or cutdown). After six cumulative attempts to

place an IV or administer ORS, two additional (20%) providers chose NG. Providers of each level of training except pediatrician (one community health worker, two nurses, five clinical officers, and two medical officers) and from each practice site were among those who chose to place an NG at any time during rehydration.

When discussing rehydration, the majority of providers first considered oral rehydration solution or intravenous fluids. "So you give ORS. That would be the first. Secondly apart from encouraging the mother to continue breastfeeding, you give ORS. Then if the child is unable to take orally, you give IV."

When providers discussed encountering difficulties, they tended toward attempting another IV, attempting an intraosseous line, or referring. "For example, if there is difficulty or the child can't retain [ORS] then I will go for IV fluids, if the difficulty is in getting an IV line then I will go for intraosseous." "If there are difficulties, you can refer."

Specific challenges regarding providing rehydration primarily included placing intravenous lines. "Mostly [the challenge is] maybe not getting the IV line. Sometimes we don't have even the appropriate cannula for children. And another possibility is that we run out of things. Another issue is parents coming when the child is severely dehydrated and accessing a vein is difficult, so that you really take a lot of time and the baby deteriorates." Other issues related to the availability of resources and ability to refer to another facility or higher level of care. "There are many challenges. One of them is putting an IV line and failing. Another one is refusal of referral. When you refer the client, the response that you will get is that they don't have the money for referral." Other perceived challenges coded from interview transcripts can be seen in Table 2.

When discussing use of nasogastric tubes for rehydration, providers discussed concerns regarding availability and knowledge. "We need availability of the NG tube itself and having the fluids available. I have not seen NG tubes in this hospital." When asked about use of NG for rehydration, one said "I don't think they (NGs) are as effective because it will depend on absorption, especially if the child has diarrhea. Then the child will have [an] absorption problem, so it's not as effective. But it's still an option, the last option." Another stated, "I am not aware if anyone is supposed to use an NG tube for rehydration." The majority of providers felt that IV fluids were better for rehydration; only thirteen providers (one community health worker, six nurses, four clinical officers, one medical officer, and one pediatrician) felt NG rehydration was as effective as IV.

The majority of providers were familiar with the guidelines and received some training during school. The familiarity and degree of training increased as provider level increased. Eleven nurses, one community health worker, and three clinical officers stated they were unfamiliar with the guidelines. Nurses were more likely to state that they had not received any training. Otherwise, the majority had received their training in school, and only a few (nine providers) had received specific WHO Integrated Management of Childhood Illness (IMCI) training. Of those with specific IMCI training, four chose NG for rehydration.

Most felt inadequately trained regarding WHO guidelines and believed more training would be beneficial to improving care for dehydrated children. "I think training would assist. Maybe we should just make it the routine to follow the procedure when we get such cases. Maybe we put some emphasis on fixing an NG tube…If you fail maybe to get the

**Table 2  Categorized codes from interview transcripts.**

**Participants' Challenges to Rehydration**

    Children presenting too late

    Difficulty placing IVs

    Difficulty referring (process, family refuses or can't afford, no ambulances)

    Not having resources

    Not having enough personnel

    Not having enough room

    Not having a rehydration center

    Parents worried child very sick when using tubes

    Difficulty with knowing amount of fluid to give

**Knowledge and beliefs about NG tubes**

*Reasons participants' believed NGs should not be used*

    NG can cause diarrhea

    Not using NG because puncture mucosa

    Not using NG due to HIV status

    Not using NG if infection

    Not using NG if patient comatose

    Not using NG if vomiting

    Not using NG when child has hole between teeth

*Beliefs about NG tubes*

    Use NG as last resort

    NG not as good as IV

    Only using NG with intestinal obstruction

    Use NG only in neonates

    Use NG for severe dehydration

    Use NG only for feeding, not rehydration

    Use NG when cannot take orally

    Use NG while struggling for IV

    Use NG with malnutrition

*Barriers to NG use*

    Forgetting about NG

    Not familiar with guidelines

    Not knowing NG sizes

    Not knowing can use NG for rehydration

    Difficulty inserting NG tubes

    Not having experience/training on NG tubes

    Not knowing how to insert NG tubes

**Participants' identified needs for improvement**

    Education for parents

    Experience/training with NG tubes

    Prevention of diarrheal illness with health education

    Needing NG integration into protocol

    Needing NG materials available

    Needing ORS/rehydration fluids

    Needing personnel (doctors, consultants, trained personnel)

    Needing updates in education

line, why not fix the NG tube. We try to do it practically [and] then we get used to [using NG tubes for rehydration].''

Providers were asked about parents' perceptions of NG tubes. ''They think if you want to insert an NG tube, this must be a very sick child. Actually before you insert an NG tube, you have to tell them why you are doing this and to reassure them there is no harm.'' Another stated, ''[For] the majority, I don't think they will mind, but occasionally a few [parents] associate tubes with bad things happening to their children and a child who is about to die. But [the] majority, I think they will cooperate.''

## DISCUSSION

This is one of the first studies to provide insight into current practice patterns for resuscitation of children with severe dehydration and the significant barriers that providers face when caring for these patients in a resource-limited setting. In general, few providers chose NG for rehydration in a case scenario when IV attempts failed. Additionally, this study provides evidence that providers lack knowledge regarding guidelines that recommend use of NG tubes as an alternative to intravenous fluids. Another barrier that was often mentioned was NG availability. This too may be linked back to lack of knowledge of NGs for rehydration; NG tubes are often available in hospitals for surgical cases, malnutrition, and feedings, but they may not be stocked for the purpose of rehydration in acute care facilities.

Providers recognized that IVs are difficult to obtain in this population and may not be readily available in many healthcare facilities in the community. With referral being difficult, other feasible and effective modalities are critical to rehydrating children. However, providers did not regard NG rehydration as equivalent to IV or consider it in place of oral rehydration therapy. Similar to our study, *Conners et al. (2000)* conducted a survey of pediatric emergency medicine fellowship directors that also found most providers believed IV to be better than oral hydration. Additionally, *Fonseca, Holdgate & Craig (2004)* found that worldwide, providers choose intravenous rehydration over enteral, including NG. This would suggest that there exists a widespread knowledge gap on the efficacy of NG rehydration for severe diarrhea and that gap may exist in both resource-rich and resource-limited regions of the world.

When considering parents' expectations for providers treating their children, providers felt parents would be hesitant about NG tubes, primarily due to lack of familiarity. *Conners et al. (2000)* also cited parents' expectations for IV over oral rehydration as a main reason for choosing IV. *Nir et al. (2013)* found that the majority of parents expected IV over oral or NG rehydration, often because of previous treatment with an IV. A study in Nigeria, however, found that most parents were accepting of insertion and use of an NG tube (*Aliyu, Abdulsalam & Teslim, 2015*). This may be due to no past experience leading to specific expectations or could be related to cultural beliefs and acceptance. For those parents that initially rejected NG placement, they were willing to accept NG for feeds with education (*Aliyu, Abdulsalam & Teslim, 2015*). While our study did not directly ask parents, providers felt that parents would be accepting of NG rehydration after education from providers.

These studies emphasize the need to adhere to evidence-based guidelines and educate providers and families. Failing to implement guidelines sets up expectations for parents that they should get an IV and fails to demonstrate the importance and efficacy of enteral hydration. If healthcare providers are not taught these rehydration guidelines and instead lean on observed common practices, they also fail to understand and utilize life-saving interventions, like NG rehydration.

Within our discussions with providers, they expressed openness to more training and felt implementing a rehydration protocol would help with NG rehydration when an IV was unobtainable. Therefore, providing more education regarding guidelines and implementing these in practice could equip providers to better care for children with dehydration and potentially improve outcomes. Ensuring education related to rehydration protocols and making NGs readily available, including in remote health posts, would equip healthcare providers at all levels to care for dehydrated children.

The study is not without limitations. The data relies on the practice of practitioners in one part of Kenya, making it difficult to generalize the results to other geographical locations. However, as mentioned above, other studies have also found limited use of NGs for rehydration (*Fonseca, Holdgate & Craig, 2004*; *Blum et al., 2011*; *Talbert et al., 2012*; *Nir et al., 2013*). Another limitation to this study is use of a case-based survey of practice patterns over observed clinical practice. We believe, however, that the survey prompts clinicians to choose another rehydration method sooner than in actuality. Despite this, clinicians still failed to choose NG rehydration according to guidelines. Also, the study did not collect years of experience for providers, which may impact their clinical practice and knowledge of guidelines. However, the majority of providers were unfamiliar with guidelines or failed to follow guidelines, emphasizing the need for further training.

## CONCLUSIONS

Healthcare providers in western Kenya do not use nasogastric tubes for rehydration according to WHO guidelines for diarrheal illness with severe dehydration. Barriers to the use of NG tubes were lack of knowledge and availability. Education and implementation of guidelines using NG tubes for rehydration may improve outcomes of children suffering from diarrheal illness with severe dehydration.

**List of abbreviations**

| | |
|---|---|
| **NG** | Nasogastric Tube |
| **WHO** | World Health Organization |
| **IV** | Intravenous |
| **ORS** | Oral rehydration salts |
| **MTRH** | Moi Teaching and Referral Hospital |
| **AMPATH** | Academic Model Providing Access to Healthcare |

## ACKNOWLEDGEMENTS

The Pediatric Project Development Team within the Indiana Clinical and Translational Science Institute also helped with the development of this project protocol.

### Funding

This study was funded by a Project Development Team within the ICTSI NIH/NCRR Grant Number TR000006. The Pediatric Project Development Team within the Indiana Clinical and Translational Science Institute also helped with the development of this project protocol.

### Grant Disclosures

The following grant information was disclosed by the authors:
Project Development Team within the ICTSI NIH/NCRR: TR000006.
The Pediatric Project Development Team within the Indiana Clinical and Translational Science Institute.

### Competing Interests

The authors declare there are no competing interests.

### Author Contributions

- Darlene R. House conceived and designed the experiments, performed the experiments, analyzed the data, contributed reagents/materials/analysis tools, wrote the paper, prepared figures and/or tables, reviewed drafts of the paper.
- Philip Cheptinga conceived and designed the experiments, performed the experiments, analyzed the data, contributed reagents/materials/analysis tools, reviewed drafts of the paper.
- Daniel E. Rusyniak and Rachel C. Vreeman conceived and designed the experiments, analyzed the data, contributed reagents/materials/analysis tools, reviewed drafts of the paper.

### Human Ethics

The following information was supplied relating to ethical approvals (i.e., approving body and any reference numbers):

The study was approved by both Indiana University Institutional Review Board (Approval IRB #1209009483) and the Institutional Research and Ethics Committee at Moi Teaching and Referral Hospital (MTRH) (Approval #000936).

### Data Availability

The interviews and datasets supporting the conclusions of this article are available in the Open Science Framework repository: https://osf.io/6zabn.

### Supplemental Information

Supplemental information for this article can be found online at http://dx.doi.org/10.7717/peerj.3829#supplemental-information.

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
