# Peer review of "Qualitative study of healthcare providers’ current practice patterns and barriers to successful rehydration for pediatric diarrheal illnesses in Kenya"

_PeerJ, doi:10.7717/peerj.3829_

## Round 0.1 · original submission · Major Revisions

In considering the responses from the two reviewers for this article I am inclined to agree with Reviewer 1 and ask for major revisions.

Reviewer 2 has rejected the manuscript on the grounds that NG tube should always be used before IV for rehydration and therefore the study is flawed from the beginning. I disagree with this and actually the IMCI tends to propose IV fluids for severe dehydration and only discusses NG fluids when IV is not possible. I therefore feel that this is an interesting study and its findings are valid.

Having said that, as reviewer 1 notes there are a number of things that need correcting and I wonder if you could specifically address the verbal vs. written consent question posed by them. It is also very important that care providers are reported by professional/training group as this makes it easier to translate any findings from this study.

Reviewer 1 ·

Basic reporting

Clear English:
Overall, the language used was clear, unambiguous, and reflects current professional standards. Some minor grammatical errors. Specifically, place a comma after using "despite" when followed by a contrasting clause (ex. lines 52 & 95). Additionally, some of language included may benefit from minor revision (examples below).

Line 110: Use of the phrase “in our experience” seems very anecdotal. Though there may be very little, if any, research regarding the use of NGs, consider including literature that looks at which rehydration practices are most used in Kenyan healthcare settings. While your experiences are more than valid, offering a more concrete reference, if available, would strengthen your point.

Line 113: “There is no literature” is a very strong statement. While I am sure your review of the available literature was exhaustive, you may want to avoid such an absolute. Consider using “based on an exhaustive (or extensive) review of the literature…” or “No literature pertaining to x was identified”.


References:
Most of your references are 6 or more years old. One is almost 30 years old. While many seem relevant to the current study, consider including literature published in the last 5 years.

Line 92: You use a CDC reference from 2011. I would encourage you to use a more current reference. There are mortality estimates--those cited in CDC reports--that have been published in the last 3 years.

Line 94: To my knowledge, cholera does not drive diarrheal rates among infants and young children in Kenya. Consider referencing the pathogen(s) that drive diarrheal rates among this population in this setting. The Global Enteric Multicenter Study has a number of published papers that could provide some additional information.


Figures and tables:
Table 1 could use some additional development. Consider including more demographic/background information in table 1, if available.

Self-contained:
Overall, the paper was strong and well-written. However, there are several areas of the paper that could benefit from additional consideration. Specifically, the methods and results sections (discussed below).

Experimental design

Original research:
The research is original and fits well within the scope of the journal.


Research question:
The research question is clear and fills a important knowledge gap in the literature.


Rigorous investigation:
Based on the information included, the research meets high technical and ethical standards. However, I do have one question regarding consent.

Line 126: Since your study population was limited to pediatric care providers, all of whom are likely literate, why did you only collect verbal consent? Verbal consent is generally reserved for studies exploring socially sensitive/taboo topics or for populations where literacy is limited. Were there barriers to collecting written consent?


Methods:
When writing up a methods section, I generally work to answer the who, when, what, where, and how of the study. While simplistic, it serves to ensure that at least the basic components of the methods section are articulated. While you clearly address the where, who, and what - where (western Kenya), who (44 healthcare providers at various facilities around MTRH and AMPATH), what (a case-scenerio and semi-structured interviews), you do not provide any information as to “when” the study was conducted, “how” participants and health facilitates were selected for inclusion or “how” much time each participant contributed to the study. Further, I do not see any reference to your sampling technique. Though based on the description offered, it seems as convenience sampling was used. This should be made explicit. Noting the sampling techniques used, even when describing a study with a small sample, provides the reader with a better frame to interpret the results. Also, please describe the number of healthcare facilities in the region and the number you planned to include in the study.

Line 148-156: The description of the study population needs to be improved. As noted, be sure to include a more developed description of sampling techniques used. As it stands, it is unclear if you selected either the research sites or the individual participants based on convenience, purposive, snowball, and/or random sampling. Though you have a small sample, this information in necessary for framing the results presented. Also, would community health workers receive training regarding NGs and/or would they typically be expected to administer them or be familiar with the process. If this is not included in their training, their responses may skew the results. Lastly, how many healthcare facilitates sit within a certain radius of MTRH and AMPATH and how many of these healthcare facilities were represented in data collection?

Validity of the findings

Once other comments are considered, you may want to update the discussion and conclusion accordingly.

Additional comments

Results:
You seemed to lump all providers together thereby failing to acknowledge the varying levels of training received. Community health workers (CHWs) receive far less training than clinicians. Moreover, given the level of training received by CHWs, should they be expected to be familiar with the administration and/or use of NGs for rehydration or even the related WHO guidelines? There are arguably 3 or 4 levels of providers with regard to training. However, the results section, as currently written, does not seem to consider the extent to which provider training may influence NG use. You essentially treat all providers as equal leaving me with several answered questions: Were specific provider types more likely to describe NGs as less effective? Were specific provider types more likely to be familiar with the WHO guidelines? Did this change according to practice type (i.e. providers at a referral hospital are more likely to use the procedure)? Other similar considerations are offered below.

Line 266: How many providers out of 44 were unfamiliar with the guidelines? Again, did you look at this according to provider type? Or practice type? Further, is it possible that the results were impacted by other factors? How long did each provider practice? How certain are you that these approaches/techniques were taught? At some point prior to the study, did all providers receiving training on the administration of NGs? Was this influenced by provider type? Additionally, nurses represented almost half of the participants included. How might this impact the study results?

Additional general comments:

Title: Rehydration can occur both within and outside of healthcare settings. Your title should reflect your focus on healthcare providers/healthcare settings.

Line 104: You suggest that NGs can be applied before arrival to the hospital. Given the settings and the more typical modes of transportation used when trying to access healthcare, who would be applying the NGs before arrival. As written, it implies that non-healthcare providers, those involved in transport, could perform the procedure.

Line 164: Consider writing out emergency department rather than using the acronym “ED”.

Line 175: Every interview is not likely to be a “key-informant” interview in the strictest interpretation of the term. I would encourage you to remove that language as it does not add anything to the sentence and may distract the reader from the main point you are trying to make.

Line 183: See comment regarding line 126.

Line 204: Be specific about the type of triangulation referenced (investigator triangulation).

Line 209: In general, you should avoid beginning a sentence with a number. Spell out the number 44.

Line 313: Consider this statement in concert with the statement regarding line 110. None the studies you allude to are clearly references above.

·

Basic reporting

The English language is good, fluent, correct and clearly understood.

Experimental design

There are 3 main problems with the study:
1. According to Good Clinical Practice (GCP) one needs an approval for a study, and informed consents, for studies dealing with patients. This study does not deal with patients: it deals with caregivers. This study is a questionnaire for caregivers. Well, it is the same as examination, so it needs neither approval, nor informed consents.
2. The sample size is 44 caregivers. The reader has no idea about the population size of caregivers out of which the sample was taken, and its distribution. It could be that the sample was too small, and hence the study may not make any conclusion regarding the population of caregivers in Kenia. Moreover, the authors did not explain how they chose the participants in the sample, so even the authors have no idea if the sample that they chose is representative.
3. According to WHO guidelines, one does NOT use NG tube after failing an attempt placing IV line; rather, one uses NG tube BEFORE even trying the first attempt placing IV line! So, the questionnare should be changed, all the study should be redone.

Validity of the findings

The findings can not be discussed, as there are the study is mal-designed.

Additional comments

I am sorry, but the manuscript is not acceptable for publishing anywhere, due to serious design mistakes.

---

## Round 0.2 · accepted · Accept

Thank you for your resubmission. All of the points in the reviewer's comments have been satisfactorily addressed.